# Analysis of Wire-Cut Electro Discharge Machining of Polymer Composite Materials

**DOI:** 10.3390/mi12050571

**Published:** 2021-05-18

**Authors:** Timur Rizovich Ablyaz, Evgeny Sergeevich Shlykov, Karim Ravilevich Muratov, Sarabjeet Singh Sidhu

**Affiliations:** 1Department of Mechanical Engineering, Perm National Research Polytechnic University, 614000 Perm, Russia; kruspert@mail.ru (E.S.S.); karimur_80@mail.ru (K.R.M.); 2Department of Mechanical Engineering, Sardar Beant Singh State University, Gurdaspur 143521, India; sarabjeetsidhu@yahoo.com

**Keywords:** wire electro-discharge machining, polymer composite materials, processing precision, interelectrode gap

## Abstract

This study presents the analysis of wire-cut electro-discharge machining (WIRE-EDM) of polymer composite material (PCM). The conductivity of the workpiece is improved by using 1 mm thick titanium plates (layers) sandwiched on the PCM. Input process parameters selected are variable voltage (50–100 V), pulse duration (5–15 μs), and pause time (10–50 μs), while the cut-width (kerf) is recognized as an output parameter. Experimentation was carried out by following the central composition design (CCD) design matrix. Analysis of variance was applied to investigate the effect of process parameters on the cut-width of the PCM parts and develop the theoretical model. The results demonstrated that voltage and pulse duration significantly affect the cut-width accuracy of PCM. Furthermore, the theoretical model of machining is developed and illustrates the efficacy within the acceptable range. Finally, it is concluded that the model is an excellent way to successfully estimate the correction factors to machine complex-shaped PCM parts.

## 1. Introduction

Recently, replacing metallic machine parts with composite material has been seen as a potential alternative to various issues, including high metal costs, rusting, and the weight of the components. In the modern machining industry, composite materials that possess similar or even enhanced physical and mechanical properties compared to metals are highly encouraged [1,2,3]. Polymer composite materials (PCMs) are recognized as a group of difficult-to-machine materials [4]. The development of light-weight PCMs plays a significant role in aviation and many critical industrial applications. These materials are economically efficient and reduce the CO_2_ emission load [1]. The binders used in PCM have good strengths and are heat resistant, resulting in high elastic strength and operational stability. In contrast, the matrix phase in the PCM is the ductile phase that transfers the external load stress to the filler phase. The filler/reinforcement used in a PCM determines its mechanical properties, such as strength, stiffness, and deformability. The filler used may be carbon/ceramic fibers. These fibers have good physical and mechanical properties. These fibers are converted into fabrics by weaving [5,6,7]. A typical PCM is shown in Figure 1a, and the weaving pattern of the fibers forms the reinforcement phase and the possible defects in conventional machining.

PCMs possess many limitations during their machining due to mismatch of properties in the filler and matrix phases [8,9]. PCMs tend to delaminate during machining due to layering, structural heterogeneity, high hardness of the filler material, and low plasticity of the binder. The machining of PCMs also results in high cutting forces and vibrations, which causes pullout of fibers and other detrimental effects at the machined zone. Delamination and pull out of fibers can be observed in Figure 1b, which diminishes the quality of the fabricated feature. Moreover, during PCM machining, a high tool wear rate causes a decrease in productivity and enhances manufacturing costs. To avoid such problems, various non-conventional machining methods have been developed for processing these materials (Figure 2). Despite having many advantages over conventional machining methods, non-conventional processes such as electrochemical and chemical processing are hazardous to the environment [10,11]. On the other hand, techniques such as laser treatment [12,13,14,15], plasma treatment [16,17], electron beam treatment [18,19], ultrasonic treatment [20,21,22], and water jet treatment [23,24,25,26,27] are considered advantageous only where accuracy is not the primary concern. Moreover, these machining methods have disadvantages such as thermal destruction of the matrix phase and lack of accuracy, limiting their application for machining small-sized components.

Among the various non-conventional machining techniques, Wire-cut Electrical Discharge machining (WIRE-EDM) has been proven to be a potential candidate to prepare small components with high accuracy [28,29,30]. In this process, thermal-electrical energy is involved in transforming the electrical energy to heat energy sufficient to melt the target zone, and an accurate curved profile can be obtained on metallic as well as polymer composite materials. A schematic diagram of the WIRE-EDM process is shown in Figure 3. In this process, the wire electrode moves vertically (mostly) over sapphire or diamond guides, which are controlled by Computer Numerical Control (CNC) program. A steady stream of deionized water or other fluid is used as a dielectric medium, flushes out debris, and cools the workpiece and the wire electrode. The dielectric fluid gets ionized, thereby producing a spark between the wire electrode tool (ET) and the workpiece electrode (WE) (Figure 3). The ionization of dielectric fluid depends on many factors of WIRE-EDM, such as properties of the working fluid, degree of contamination of the working fluid with erosion, the material of the electrodes, and dielectric flow pressure. The amount of thermal energy generated within the electrodes affects the amount of material removed from the surfaces of the ET and WE differently. This unevenness in material removal depends on the thermophysical properties of the ET and WE, and the process parameters of WIRE-EDM [29,30,31]. By varying these factors, electrode erosion can be precisely controlled. The spark energy within the electrodes depends on the voltage, the pulse formation time, the state of the working fluid, and the size of the interelectrode gap. Thus, the accuracy of the WIRE-EDM of PCMs is influenced by the size of the inter-electrode gap. The error of the inter-electrode gap depends on the inhomogeneity in the structures/properties of ET and the WE, and also the properties of the working fluid. The WIRE-EDM of PCMs results in the formation of dimples/craters of different sizes on the surface of the workpiece. These randomly formed dimples/craters are a factor that complicates the prediction of the inter-electrode gap [31].

The application of WIRE-EDM on PCMs has been investigated by various researchers [8,9]. However, it is a known fact that the conductivity of PCMs is limited. Thus, during the WIRE-EDM process, the resin of the PCM was destroyed at the edges of the holes. This was due to high temperatures and ineffective cooling at the machining zone. 

Abdallah et al. [32] used WIRE-EDM to study the effects of gap voltage, current, pulse-on time, and pulse-off time on the material removal rate (MRR), top and bottom cut-width (kerf), and workpiece edge damage in unidirectional carbon fiber reinforced polymer (CFRP) composites. Current and pulse-off time were found to be statistically important parameters in terms of MRR, with current being the only factor affecting cut-width on the top surface. Recently, Dutta et al. [33] investigated a modified version of WIRE-EDM for CFRP composite cutting by using H13 steel plates as sandwich, assisting the electrodes to trigger the electrical spark during CFRP composite WIRE-EDM. Using metal plates (H13 steel) as assisting electrodes, problems such as incomplete cuts and deviations in the machining direction during CFRP WIRE-EDM were controlled. The results showed that increasing the current (from 2A to 12A) reduced the cutting time (by 60.95%) while keeping all other parameters constant.

Likewise, in similar studies [34,35,36,37,38,39,40,41,42,43,44,45,46,47,48] related to the WIRE-EDM of PCMs, it was observed that the quality and accuracy of holes in a low-conductive material can be regulated by applying a conductive layer above the non-conductive PCM. Also, the development of theoretical models of the WIRE-EDM of PCMs provides a guide to obtain the accuracy required in the process [37].

A schematic of WIRE-EDM process is represented in Figure 4. Herein, the size of the ET (2R), and the value of interelectrode gap(overcut)S are taken into account for accurate machining of the product. The correction in machining can be done in reference to the center of the ET(wire)in a CNC controller. 

The material removal rate from the workpiece during a single pulse is estimated by the following equation [49,50,51]:(1)$$MRR=\frac{m}{t_{on}}$$
where $t_{on}$ is the duration of a single pulse (μs), and *m* is the weight lost during the EDM process (kg). Data in the literature [31,51] indicate that *MRR* depends on the value of the interelectrode gap *S* (m), the feed rate *V* (m/s), the physical and mechanical properties of the processed material, and the workpiece thickness *h* (m). *MRR* is calculated using Equation (2):(2)MRR=2(R+S)hVρ
where *R* is the radius of the ET (m), *S* is the interelectrode gap (m); *ρ* is the density of the processed material (kg/m^3^), *h* is the thickness of workpiece (m), and *V* is the feed rate. 

The spark energy, *W* (J), is released in the interelectrode gap and is distributed between the ET and the workpiece. The material is removed from the workpiece by the mean of the spark energy [31]. The pulse energy is calculated as:(3)$$W=\int_{0}^{t_{p}}UIdt_{on}$$
where *U* is the voltage, (V); *I* is the current strength, (A); *t_on_* is the pulse duration, (μs); and *tp* is the pulse width (μs). A correction factor is introduced in Equation (3) to improve the accuracy of the calculations.
(4)$$W=\left(\eta_{u}\right)UIt_{on}$$

The coefficient for a fraction of the energy ($\eta_{u}$) utilized in the machining process is represented in Equation (5) [31,50]:(5)$$\eta_{u}=\left(1-K_{1\;}\right)\left(1-K_{2}\right)$$
where *K*_1_ is the amount of energy lost during the heating and evaporation of the dielectric fluid; *K*_2_ is amount of energy lost in the ET. The amount of heat *Q* (J) transferred to the WE for heating and melting is determined by the formula:(6)$$Q=m\left(C_{1}\Delta T_{1}+\lambda +C_{2}\Delta T_{2}+r\right)$$
where *m* is the mass of the workpiece (kg); *C*_1_ is the specific heat capacity of the material in the solid state (J/kg K) *C*_2_ is the specific heat capacity of the material in the liquid state (J/kg K); ΔT is the temperature difference between the initial and final heating points (K); *λ* is the specific heat of fusion of the material (J/kg); and *r* is the specific heat of vaporization (J/kg).

Taking into the account the equal coefficients of energy loss in Equation (4) and Equation (6) (i.e., *W* and *Q*) and $Z=C_{1}\Delta T_{1}+\lambda +C_{2}\Delta T_{2}+r$. Equation (1) is represented as: (7)$$MRR=\frac{Q\eta_{u}\mathrm{UI}}{ZW}=\frac{\eta_{u}UI}{Z}$$
thus equating expressions (2) and (7). The value of the interelectrode gap *S* is calculated as:(8)$$S=\frac{\eta_{u}UI}{2ZhV\rho }-R$$

The value of B (as shown in Figure 4) can be calculated, and thus the cut width (L) is obtained. This theoretical model can be used to estimate the process parameter affecting the cut width (L), and can thus suggest the amount of correction required for accurate machining. 

The PCM is made up of both electrically conductive carbon fiber and non-conductive epoxy resin. As a result, machining such composites with prominent machining techniques i.e., WIRE-EDM or EDM is a difficult job. In the literature, research on the WIRE-EDM of PCM is limited to where the effect of WIRE-EDM on PCM is explored for higher machining rate, electrode wear, and performance in the linear cut. 

In this work, the authors investigated the performance of WIRE-EDM on a patent carbon fiber-reinforced PCM possibly adopted in the aviation industry. The voltage, pulse duration, and pause time were selected as process parameters. These parameters were statistically evaluated, and the level of the significance of factors affecting the cut width was determined using analysis of variance. Finally, the experimental values obtained for cut width were modeled mathematically in terms of significant factors using response surface methodology.

### Purpose of Study

To assess the influence of key process parameters on the cut width (kerf) and surface quality of PCM sandwiched in Titanium alloy.To develop a regression model using response surface methodology, which is further examined with the experimental results for non-linear machining cut-width on the selected PCM.To determine the trajectory of ET to machine PCM in the form of a complex shaped part, such as a gear.

## 2. Material and Methods 

### 2.1. Material

In this study, a polymer composite material (VKU-39, pl. refer https://viam.ru, 11 May 2021) used in the aviation industry was chosen. The workpiece is a laminated fibrous polymer composite made of carbon fiber twill as reinforcement/filler, with epoxy as a binder material. The property of the selected PCM is as shown in Table 1.

A PCM plate of thickness 2 mm was used for the WIRE-EDM experiments. To improve the conductivity of the PCM, a conductive layer of titanium (1 mm) was applied on both sides (Figure 5). The study was carried out on a wire-cut EDM machine, “Electronica EcoCut.” The electrode tool (ET) used was a brass wire with a diameter of 0.25 mm. Distilled water was used as a dielectric medium, and was sprayed on the ET (Figure 5) instead of immersing the PCMs in a water bath, as PCM immersion in a water bath causes defects such as swelling and filling of water.

### 2.2. Method

The process parameters selected were U—voltage, V; Ton—pulse duration, μs; and Toff—the pause time in pulse duration. The experimental runs were carried out by employing the orthogonal central composition design (CCD) matrix, where α denotes the distance between the star point and central point i.e., value of α = 1.215. The design matrix in the study was obtained with the assistance of Design-Expert software. Orthogonality in the design assisted in estimating the independent regression coefficients [49,52]. The process parameters are presented in Table 2. The output parameter was the value of the EDM cut width (L). The cut width was the wire diameter of the ET and the size of the side clearance (overcut).

The experiment design matrix and response (cut width) is presented in Table 3. Each experimental run was replicated thrice for more accuracy. A pictorial view of the machining process is shown in Figure 6.

## 3. Results and Discussion

Analysis of variance (ANOVA) is a commonly used statistical technique to test the significance of a model and the contribution of each process parameter on the experimental response. The mathematical model was predicted using Design Expert software and is summarized in Table 4.

From Table 4, it is observed that the value of adjusted R-square and predicted R-square values were higher for the 2-way interaction model. Thus, this model is suggested for further analysis.

Statistical analysis of the experimental data (Table 5) revealed the significance of the process parameters, namely voltage, pulse duration, and pause time, on the measured response i.e., cut width. The results are summarized in Table 5, with a 95% confidence level.

The 2-way interaction model F-value of 6.71 implies that the model is significant, and only 0.85% chance of noise exists in this model. From Table 5, the *p*-value less than 0.05 indicated that the model terms of voltage, Ton, and their interaction (i.e., U × T-on) were significant, whereas Toff, U × Toff and Ton × Ton were insignificant. Eliminating insignificant terms results in the improvement in the model, as presented in Table 6.

The model regression statistics for the selected model demonstrated a high R^2^ value (i.e., 0.80), which is acceptable. The predicted R^2^ of 0.6677 was in reasonable agreement with the adjusted R^2^ of 0.7504; i.e., the difference was less than 0.2.

In order to check the adequacy of model, the predicted value and the actual experimental values were compared along a 45° line, as shown in Figure 7. This implies that the proposed model is adequate and there is no violation of the independent or constant variance assumptions. 

The final equation obtained for the prediction of cut-width (L) is represented as Equation (9): (9)L=329.6−0.9475×U−6.45×Ton+0.141×U×Ton

The regression equation analysis (9) shows the combination (interaction) of input process parameters (factors) affecting the value of the cut width. 

Figure 8 and Figure 9 presents the response surface describing the dependence of the cut width on the voltage and pulse duration. 

### Examination of the Predicted Model forWIRE-EDM onComplex-Shaped PCM Parts

It is depicted from the plots (Figure 8 and Figure 9) that cut width is directly proportional to the voltage and pulse duration. Thus, for machining complex PCM parts, a high value for WIRE-EDM i.e., U = 100 V, Ton = 15 μs, Toff = 30 μs was selected. The obtained cut width path is presented in Figure 10. The cut width (Figure 10) was measured at various WIRE-EDM zones, i.e., at the entrance to the PCM, at the corner, and the end of processing. The average value of the cut width was calculated as L = 330 µm.

Figure 11a,b reveals the surface the cutting edge achieved after applying the conductive Ti-alloy layer (plates) to the surface of PCM. It is found that this method of machining results in attaining a defect-free smooth surface on both sides of the processed PCM sheet at the entrance, at the corner, and at the end.

Thus, the Ti layer sandwich method for PCM enhanced the quality of the machined surface. Additionally, it has the potential to obtain a smooth, defect-free surface within the processed slot without causing any damage to the fibers/binder of the PCM material.

The percentage error between the values of the cut-width was calculated using Equation (10). The expected value from Equation (9) was calculated as 349.6µm, and the experimental value (actual value) measured was 330 µm (Figure 10). Thus, the percentage error obtained is calculated as 5.906%, which is acceptable and shows the competency of the model.
(10)$$\%\;error=\frac{\left|Expected\;value-Actual\;value\right|}{Expected\;value}$$

The regression Equation (9) was further examined to accurately machine a gear-shaped PCM part. An “Electronica EcoCut” CNC WIRE-EDM machine was programmed. The program was used to machine a PCM workpiece into the gear shape. The machining parameters were carried out at Ton = 15µs, Toff = 30 μs, and U = 100 V. Based on this regression model (Equation (9)), the trajectory correction value was calculated as B = 0.165 mm. When machining the PCM, the offset was added into the control program using the command “G41 B” = 0.165. In Figure 12, the trajectory of the ET and the finished product “gear” are presented.

## 4. Conclusions

The experimental work reveals the dependence between the WIRE-EDM cut width and processing parameters such as voltage, pulse-on time, pulse-off time. These results can be expedited to adjust the size of the ET and ensure precision in the WIRE-EDM of PCM (VKU-39) workpieces. It is ascertained that voltage and pulse duration and their interaction are the significant factors affecting the process parameters for machining the PCM workpiece. Furthermore, a2-way interaction model is developed to estimate the cut-width, which shows excellent adequacy with the experimental values obtained. Based on the developed model, the cut-width correction factor for the trajectory of ET was estimated in a WIRE-EDM CNC machine for the accurate machining of a complex-shaped PCM product. Consequently, it is suggested that the proposed model successfully facilitates the forecasting of WIRE-EDM accuracy.

## Figures and Tables

**Figure 1 micromachines-12-00571-f001:**
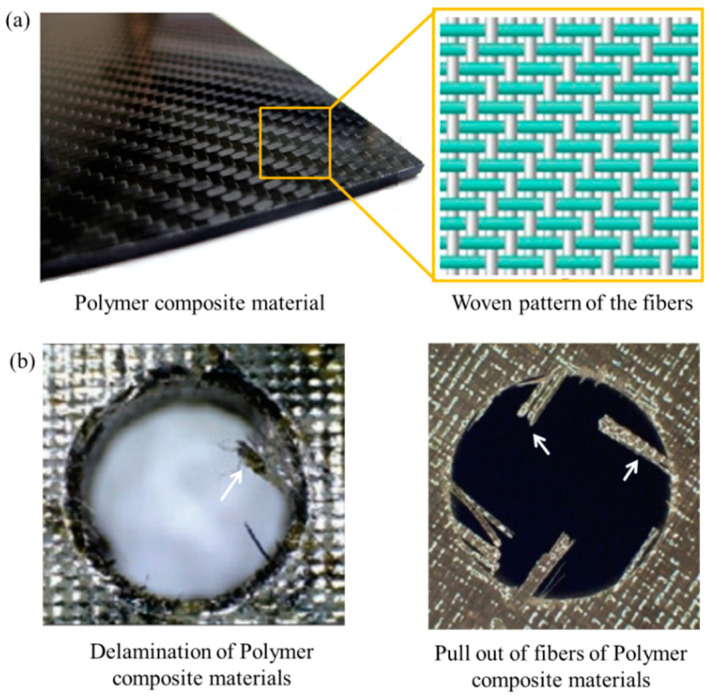
(**a**)Polymer composite materials and woven patterns of fibers of the reinforcement phase; and (**b**) delamination and pull out of fibers during the drilling of polymer composite materials.

**Figure 2 micromachines-12-00571-f002:**
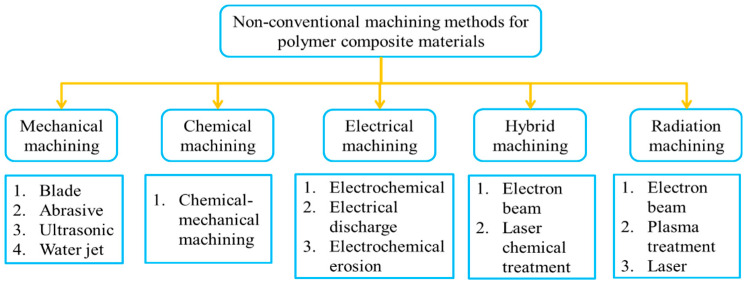
Non-conventional machining methods of polymer composite materials.

**Figure 3 micromachines-12-00571-f003:**
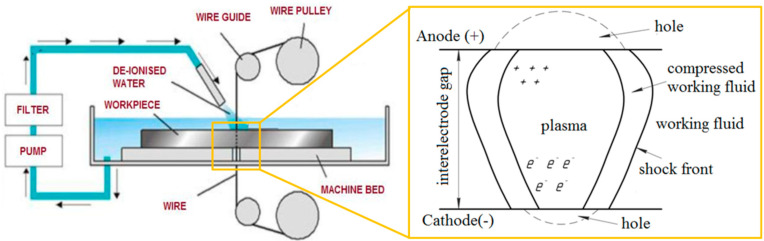
Schematic diagram and working principle of the wire-cut electro-discharge machining (WIRE-EDM) process [31].

**Figure 4 micromachines-12-00571-f004:**
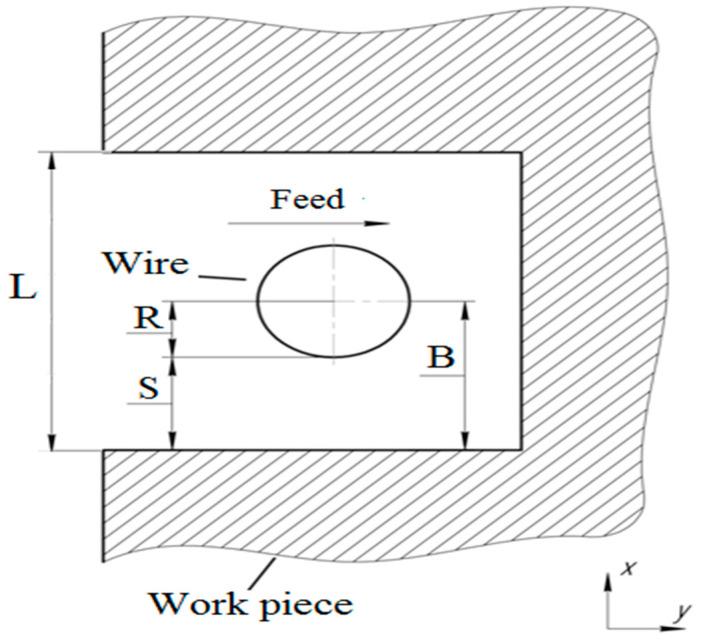
WIRE-EDM processing in the XY plane: R—radius of the electrode tool (ET); S—interelectrode gap; (B)—reference for correction; L—cut-width (Kerf).

**Figure 5 micromachines-12-00571-f005:**
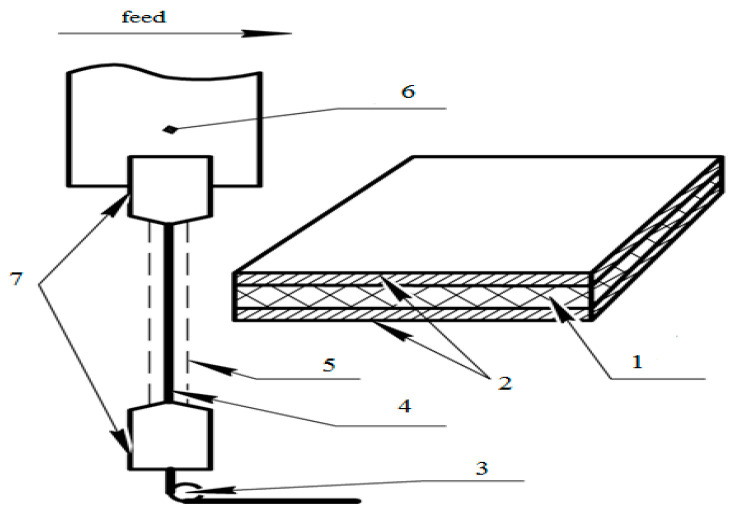
Schematic of the WIRE-EDM of polymer composite material (PCM) with conductive layers used in the present study (1: PCM, 2: conductive Ti-layers, 3: tension roller, 4: ET wire, 5: flushing, 6: gearbox, 7: upper and lower diamond wire guides).

**Figure 6 micromachines-12-00571-f006:**
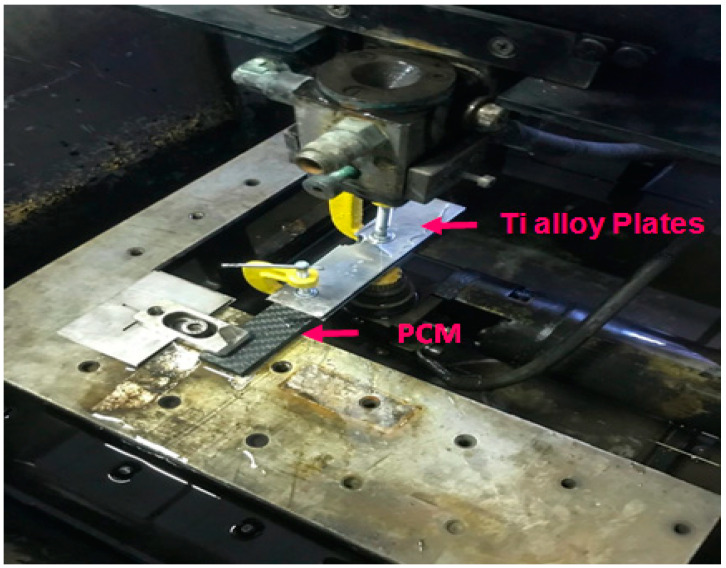
Pictorial view of PCM machining setup.

**Figure 7 micromachines-12-00571-f007:**
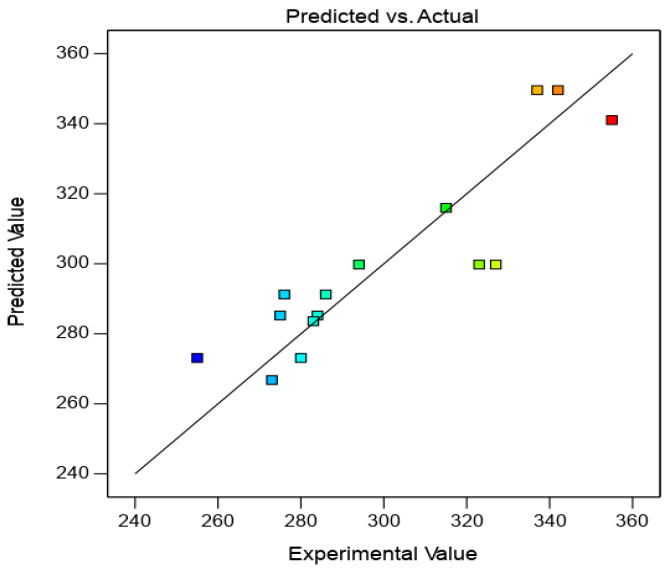
Comparison of model-predicted values versus actual experiment values for cut width.

**Figure 8 micromachines-12-00571-f008:**
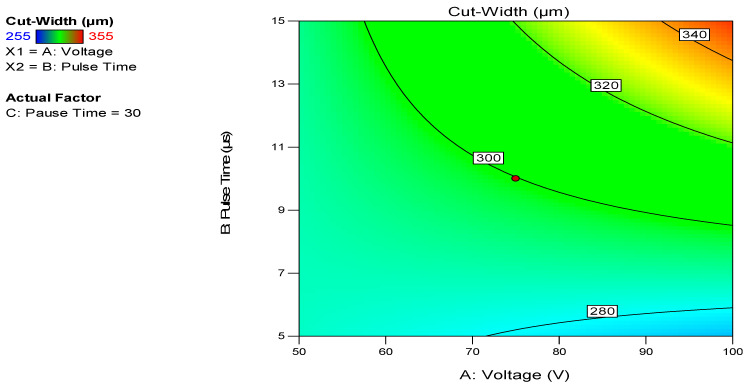
Contour plot for cut-width dependance on voltage and pulse duration for pause time Toff = 30 µs.

**Figure 9 micromachines-12-00571-f009:**
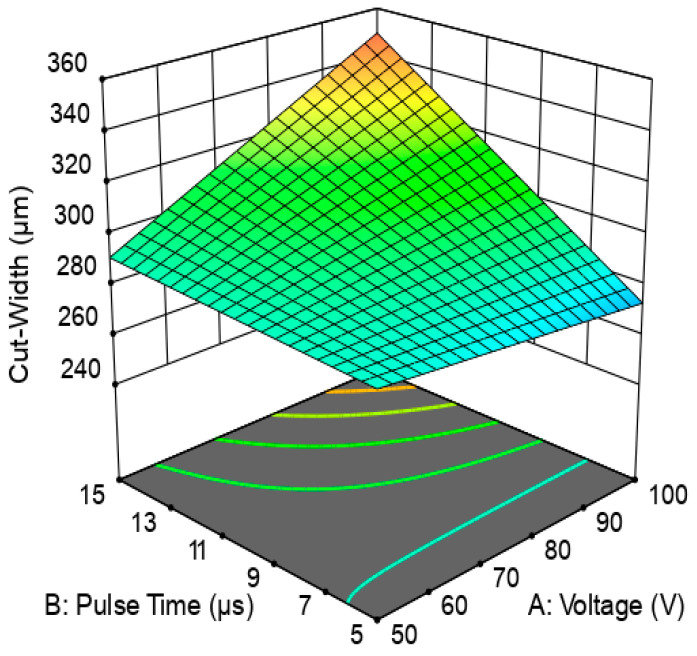
Response surface 3D-plot for cut-width dependence on voltage and pulse duration for pause time Toff = 30µs.

**Figure 10 micromachines-12-00571-f010:**
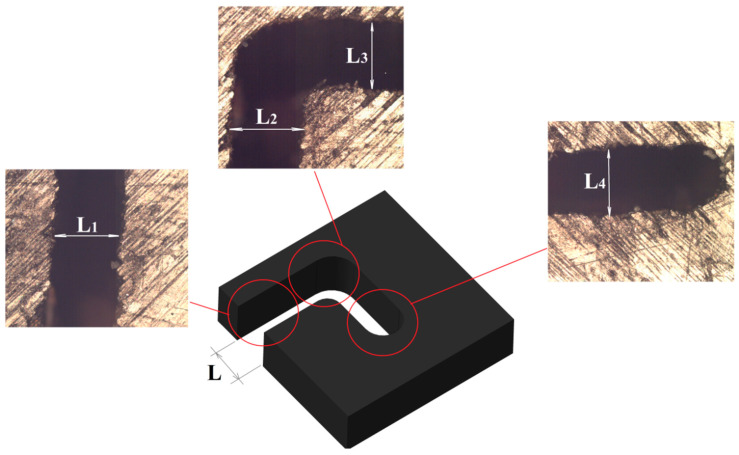
Values of the cut width for PCM wire-cut electro-discharge machining (WIRE-EDM) at U = 100 V, Ton = 15 μs, Toff = 30 μs: L_1_ = 325 µm, L_2_ = 345 µm, L_3_ = 325 µm, L_4_ = 330 µm.

**Figure 11 micromachines-12-00571-f011:**
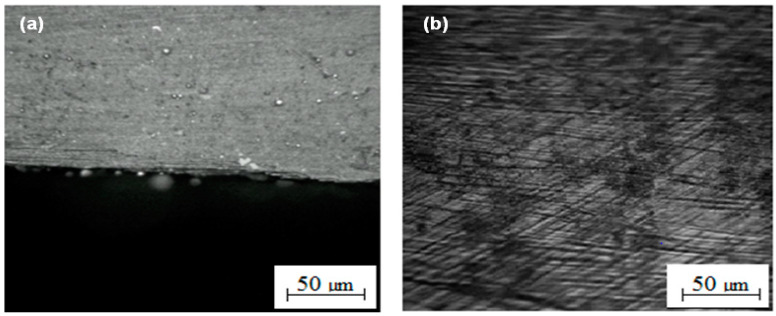
Machined PCM surfaces after application of a conductive Ti-layer on the surface of PCM; (**a**)—sheet surface at the processing zone (at the edge), (**b**)—along the cross-section of sheet width.

**Figure 12 micromachines-12-00571-f012:**
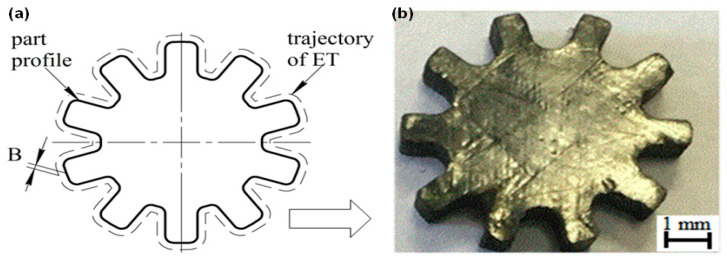
(**a**) Trajectory of the ET, and (**b**) finished product.

**Table 1 micromachines-12-00571-t001:** Properties of polymer composite material (PCM).

Property	Average Value
Filler material	Carbon fabrics twill (Porcher-3692)
Heat resistance °C	170°
Monolayer thickness, mm	0.2
Tensile strength, MPA	945
Tensile modulus, GPA	69
Compressive strength, MPA	610
Compression modulus, GPA	54
Density of carbon fiber reinforced polymer (CFRP), kg/m^3^	1550
Weaving type	Twill, at an angle of 90°
Electrical conductivity, S/m	10^−7^

**Table 2 micromachines-12-00571-t002:** Process parameters of wire-electrical discharge machining (EDM).

Factors, Units	Units	Lower Level(−1)	Upper Level(+1)	Average Level	Lower “Star” Point	Top “Star” Point
U	V	50	100	75	40	110
Ton	µs	5	15	10	2	20
Toff	µs	10	50	30	5	60

**Table 3 micromachines-12-00571-t003:** Experimental central composition design (CCD) matrix with the responses the experiment obtained.

Exp. No.	Process Parameters			Response
	Voltage U (V)	Pulse Duration Ton (µs)	Pause Time Toff (µs)	Cut-Width L (µm)
1	100	15	50	337
2	50	15	10	286
3	100	5	50	280
4	75	10	60	323
5	75	10	5	327
6	50	5	10	275
7	50	5	50	284
8	75	20	30	355
9	100	15	10	342
10	50	15	50	276
11	75	2	30	273
12	110	10	30	315
13	75	10	30	294
14	100	5	10	255
15	40	10	30	283

**Table 4 micromachines-12-00571-t004:** Mathematical model analysis.

Model	*p*-Value	Adjusted R^2^	Predicted R^2^	
Linear	0.0127	0.5061	0.2412	-
2-way interaction	0.0663	**0.7100**	**0.3711**	Suggested Model
Quadratic	0.4820	0.7049	−0.0121	-

**Table 5 micromachines-12-00571-t005:** Analysis of variance (ANOVA) of process parameters for 2-way interaction model.

ANOVA for Response Surface						
Source	Sum of Squares	df	Mean Square	F-Value	*p*-Value	
Model	10,655.21	6	1775.87	6.71	0.0085	Significant
U-Voltage	1593.02	1	1593.02	6.02	0.0397	-
Ton	6189.06	1	6189.06	23.40	0.0013	-
Toff	33.21	1	33.21	0.1256	0.7322	-
U × Ton	2485.13	1	2485.13	9.40	0.0155	-
U × Toff	55.13	1	55.13	0.2084	0.6602	-
Ton × Ton	300.12	1	300.12	1.13	0.3179	-
Residual	2116.12	8	264.52	-	-	-
Total	12,771.33	14	1775.87	-	-	-

**Table 6 micromachines-12-00571-t006:** ANOVA for the reduced 2-way interaction model.

ANOVA for Response Surface						
Source	Sum of Squares	df	Mean Square	F-Value	*p*-Value	
Model	10,266.75	3	3422.25	15.03	0.0003	Significant
U-Voltage	1593.02	1	1593.02	7.00	0.0228	-
Ton	6188.60	1	6188.60	27.18	0.0003	-
U × Ton	2485.13	1	2485.13	10.91	0.0070	-
Residual	2504.58	11	227.69	-	-	-
Total	12,771.33	14	3422.25	-	-	-

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
