# Peer review of "Analysis of Wire-Cut Electro Discharge Machining of Polymer Composite Materials"

_micromachines, 2021, doi:10.3390/mi12050571_

Round 1
Reviewer 1 Report
The reviewer comments of the paper «Ensuring the accuracy of wire-cut electro discharge machining of polymer composite materials»- Reviewer
The authors presented an article «Ensuring the accuracy of wire-cut electro discharge machining of polymer composite materials». However, there are several points in the article that require further explanation.
Comment 1:
The abstract needs to be improved.
In general, the abstract needs to be rewritten in more detail and concrete. Add some urgency to the stock material. What products are used. What methods are used in the article? What are the quantitative and qualitative results obtained? What a scientific novelty and practical significance.
Title's articles also need to be rewritten. Title should reflect the purpose of the article, taking into account the main research method, material, method.
Comment 2:
Introduction
The title says about accuracy, you need to add a paragraph with the analysis of accuracy with WEDM, especially for PCM`s.
Before formulating the purpose of the article, you need to add a small paragraph where you clearly show the "white" spots. What has not been studied before these studies and will be discussed in the article. What is the relevance of your research? What is the novelty? What research methods are used in the article and why? After the purpose of the article, describe briefly what has been done in each section.
Comment 3:
- Material and Methods
Write in more detail the physical and mechanical properties of the workpiece material. What is the conductivity of this material? What hardness? All this needs to be shown in the table. How are the reinforcing fibers located? Which direction? How important is the placement of these reinforcing fibers on the accuracy achieved?
Are all the formulas in the article original? If not needed appropriate citations.
What is the originality of the above mathematical formulas? What exactly did the authors do here?
The quality and resolution of all the figures needs improvement.
Are all figures original? If not, then you need appropriate citations and publisher permissions.
Specify for all machine, devices and measuring installations (Manufacturer, city, country).
The title says that accuracy is investigated. Therefore, it is necessary to clearly describe and show on the figure exactly which accuracy parameters are being investigated: surface roughness, form error, etc. How is this evaluated using mathematical models?
Comment 4:
- Results and Discussion
In Figure 9, you need to add a scale bar.
Small paragraph to figure 10 with a brief description of the conductive coating. Now the details are absolutely incomprehensible. How is the coating applied? Which setting? With what modes, thickness? How will this coating affect the further performance of the resulting product? All this should be clearly, logically described and researched.
How are the results in figures 11 and 12 related to accuracy? What parameters exactly? Figure 12 is redrawn in color.
Use a single font style and scale of figures for all articles.
The authors conclude:
"It was observed that the difference in the values obtained from the theoretical model and the regression model are no more than 15%, thus shows the competence of these models."
However, how does this 15% relate to the accuracy specified by the drawing and by the designer? For example, if the size tolerance is 25 microns, and your model gives a 15% error of 100 microns, then there will be no satisfactory model accuracy.
Figure 13. Explain in more detail what kind of gear you received? What geometry? Which tooth profile (straight-sided, involute, etc.?) For which application? What is the specified degree of precision of the gear?
Comment 5:
The conclusions also require improvement in line with the comments above.
What is the novelty of the article? What is the practical significance? What are the differences from previous works? Provide quantitative and qualitative conclusions for each parameter under study. Conclusions should reflect the purpose of the article.
Comment 6:
Careful proofreading of the English must be done.
The article has potential, but should be substantially revised and improved. Authors should carefully study the comments and make improvements to the article step by step. Mark all changes in color. After major changes can an article be considered for publication in the "Micromachines".
Author Response
The authors presented an article «Ensuring the accuracy of wire-cut electro discharge machining of polymer composite materials». However, there are several points in the article that require further explanation.

Reviewer 2 Report
The paper deals with the ensuring the accuracy of wire-cut electro discharge
machining of polymer composite materials.
The reviewer believes that the paper addresses a burning issue in manufacturing engineering,
such as improving the quality of WIRE EDM processing.
Because this approach is innovative,
the reviewer suggests the paper for publication.
But, there are needs major revision notes
that are highlighted for the best presentation of the paper.
Comment 1
Page 1 Abstract
The authors must give more details in the Abstract (add more details for description and the paper results).
Comment 2
References
The authors must format the text according to the journal's instructions.
Volume Italics
Comment 3
Page 1
polymer composite materials
(PCMs),
replace
Polymer Composite Materials
(PCMs),
Comment 4
Page 1
Figure 1 (a) along with
change
Figure 1 (a) (no bold)
Comment 5
Page 2
from Figure 1 (b), which
(Figure 2).
Change
Figure 1 (b) (no bold)
(Figure 2). (no bold)
Comment 6
Page 3
wire-cut electrical discharge machining (WIRE-EDM )
replace
WIRE-cut Electrical Discharge Machining (WIRE-EDM)
Comment 7
Page 3
Figure 3. In this
Change
Figure 3 (No bold)
Comment 8
Page 3
workpiecec
replace
workpiece
The authors must check the text for spelling and typography errors.
Comment 9
Page 3
between the wire electrode tool (ET) and
workpiece elctrode (WE) (Figure 3).
replace
between the Wire Electrode tool (ET) and
Workpiece Electrode (WE) (Figure 3).
Figure 3 (no bold)
Comment 10
The authors must delete the extra spaces into the text.
Comment 11
Page 3
accuracy of WIRE-EDM of PCMs is
of (no bold)
Comment 12
Page 4
with epoxy as as a binder material.
Delete the extra as
Comment 13
Page 4
The authors must give more details for the polymer composite material (VKU-39)
(thickness for PCM and conductive Ti- layers).
Add a table with the mechanical properties of the polymer composite material (VKU-39)
Comment 14
Page 4
(Figure 4).
No bold
Comment 15
Page 4
central composition design (CCD)
replace
Central Composition Design (CCD)
Comment 16
Page 4
Add a Figure with the experimental device (EDM machine with a typical workpiece)
Comment 17
Page 5
Table 2
The authors must explain with more details the Table 2.
What are the X0, X1 (U,V), X2 (Ton, µs), Ð¥3 (Toff, µs), Ð¥1Ð¥2, Ð¥1Ð¥3, Ð¥2Ð¥3, Ð¥'4, X'5 and X'6?
The significance of the coefficients was assessed using the Student's test.
What is the Student's test?
The authors must explain with more details.
Comment 18
Page 5
Delete the space in the page end
Comment 19
Page 6
Figure. 8. WIRE-EDM
The authors must check (From Figure 4 to Figure 8)
All Figures, Schemes and Tables should be inserted into the main text close to their first citation and must
be numbered following their number of appearance (Figure 1, Scheme I, Figure 2, Scheme II, Table 1, etc.).
replace
Figure 5. WIRE-EDM
The authors must check if the corners (in the right side of the Figure: above and down)
must have curves.
Comment 20
Page 7
ρ is the density of the processed
material (kg / m3),
How is the value of ρ in this case?
Comment 21
Page 7
width(μs).
replace (insert a space)
width (μs).
Comment 22
Page 7
. The coefficient
replace
The coefficient
Comment 22
Page 7
kg • K); С2 - specific heat capacity of the material in the liquid state (J / kg • K);
Very big •
Comment 23
The numbering of the equations should be at the end of the line on the right side of the text.
Comment 24
Page 8
Figure 9. Values
replace
Figure 6. Values
Comment 25
Page 8
and at the end .
replace
and at the end.
Comment 26
Page 8
Figure 10. Machined
replace
Figure 7. Machined
Comment 27
Page 9
Figure 11. Surface
replace
Figure 8. Surface
Comment 28
Page 9
Figure 12. Comparison
replace
Figure 9. Comparison
The authors must give more details for Figure 9 (Ton, Toff)
Comment 29
The authors must format all text according to the journal's instructions.
Page 10
Based on this theoretical model
Delete the space before the word Based.
Comment 30
Page 10
"gear" are presented..
replace
"gear" are presented.
Comment 31
Page 10
Figure 13. a - Trajectory
replace
Figure 10. a - Trajectory
The authors must measured the dimensions of the product gear, so they can insert a table or a statement to compare
the theoretical part profile and the measurement part profile.
Comment 32
Page 10
Ton = 5 µs,and Toff=10 µs.
replace
Ton = 5 µs and Toff=10 µs.
Comment 33
Increase the number of the reference papers including (primarily) from Materials.
The authors use 0 paper from Micromachines journal / 0 papers from MDPI Journals / 45 papers from journals (References)
Τhe number for papers from MDPI journals
is considered insufficient (in reviewer's opinion).
Comment 34
Improove the quality of the reference papers.
More related literature should be included in introduction section about the machining of FRPs.
Please consider the following ones, which are all related with FRPs EDM machining:
(a) DOI: 10.1504/IJMMM.2016.077712
(b) doi.org/10.3390/mi11050469
Author Response
The paper deals with the ensuring the accuracy of wire-cut electro discharge machining of polymer composite materials. The reviewer believes that the paper addresses a burning issue in manufacturing engineering, such as improving the quality of WIRE EDM processing.
Because this approach is innovative, the reviewer suggests the paper for publication. But, there are needs major revision notes that are highlighted for the best presentation of the paper.

Round 2
Reviewer 1 Report
The authors have improved the article. However, some comments were not sufficiently worked out or were completely ignored.
Therefore, it is necessary to invite the authors to re-revise the article with detailed answers to each individual comment.
1. Introduction
The title says about accuracy, you need to add a paragraph with the analysis of accuracy with WEDM, especially for PCM`s.
Before formulating the purpose of the article, you need to add a small paragraph where you clearly show the "white" spots. What has not been studied before these studies and will be discussed in the article. What is the relevance of your research? What is the novelty? What research methods are used in the article and why? After the purpose of the article, describe briefly what has been done in each section.
2. What is the originality of the above mathematical formulas? What exactly did the authors do here?
3. The title says that accuracy is investigated. Therefore, it is necessary to clearly describe and show on the figure exactly which accuracy parameters are being investigated: surface roughness, form error, etc. How is this evaluated using mathematical models?
4. Small paragraph to figure 10 with a brief description of the conductive coating. Now the details are absolutely incomprehensible. How is the coating applied? Which setting? With what modes, thickness? How will this coating affect the further performance of the resulting product? All this should be clearly, logically described and researched.
5. The authors conclude:
"It was observed that the difference in the values ​​obtained from the theoretical model and the regression model are no more than 15%, thus shows the competence of these models."
However, how does this 15% relate to the accuracy specified by the drawing and by the designer? For example, if the size tolerance is 25 microns, and your model gives a 15% error of 100 microns, then there will be no satisfactory model accuracy.
Figure 11. Explain in more detail what kind of gear you received? What geometry? Which tooth profile (straight-sided, involute, etc.?) For which application? What is the specified degree of precision of the gear?
I emphasize that the authors are expected to provide clear and detailed answers to each comment. Appropriate descriptions should be added in the text of the article. Without these changes, the article cannot be accepted for publication.
Author Response
Comment 1:
Introduction
The title says about accuracy, you need to add a paragraph with the analysis of accuracy with WEDM, especially for PCM`s.
Response: The title of the manuscript is modified as “Analysis of wire-cut electro discharge machining of polymer composite materials”. Few more reference is added in the introduction section highlighting the performance evaluation of Wire-EDM in PCMs in terms of MRR, accuracy etc. The content added is stated as:
Abdallah et al. [32] used WEDM to study the effects of gap voltage, current, pulse-on time, and pulse-off time on the material removal rate (MRR), top and bottom cut-width (kerf), and workpiece edge damage in unidirectional CFRP composites. Current and pulse-off time were found to be statistically important parameters in terms of MRR, with current being the only factor affecting cut-width on the top surface. Recently, Dutta et al. [33] investigated a modified version of WEDM for CFRP composite cutting by using H13 steel plates as sandwich assisting electrodes to trigger the electrical spark during CFRP composite WEDM. Using metal plates (H13 steel) as assisting electrodes, problems such as incomplete cuts and deviations in the machining direction during CFRP WEDM were controlled. The results showed that increasing the current (from 2A to 12A) reduced the cutting time (by 60.95%) while keeping all other parameters constant. Likewise, in similar studies [34-48] related to the WIRE-EDM of PCMs, it was observed that the quality and accuracy of holes in a low-conductive material can be regulated by applying a conductive layer above the non-conductive PCM. Also, the development of theoretical models of WIRE-EDM of PCMs provides a guide to obtain required accuracy in the process [37].
Comment 2:
Before formulating the purpose of the article, you need to add a small paragraph where you clearly show the "white" spots. What has not been studied before these studies and will be discussed in the article. What is the relevance of your research? What is the novelty? What research methods are used in the article and why? After the purpose of the article, describe briefly what has been done in each section.
Response: The composite (PCM) selected in this study is the authors patent material and recommended for use in aviation industries or special purpose medical equipment.
In the literature survey, the Wire-EDM was performed on various PCM. However, in this study, the PCM sandwiched by 1mm Titanium layer is evaluated for Wire-EDM, reflecting the novelty of this study.
The purpose is highlighted in the manuscript as:
In this work, authors aimed to investigate the performance of WIRE-EDM on the patent carbon fiber reinforced PCM, possibly adopted in the aviation industries. The voltage and pulse duration and pause time is selected as process parameters. These parameters are statistically evaluated, and the level of significance of factors affecting the cut-width is determined using analysis of variance. Finally, the experimental values obtained for cut-width are modeled mathematically in terms of significant factors using response surface methodology.
Purpose of study
To assess the influence of key process parameters on cut-width (Kerf), surface quality of PCM sandwich in Titanium alloy.
To develop the regression model using the response surface methodology, which is further examined with the experimental results for non-linear machining cut-width on the selected PCM.
Finally, determining the trajectory of ET to machine PCM in the form of complex shaped part such as gear.
- 2. What is the originality of the above mathematical formulas? What exactly did the authors do here?
Response: The mathematical model represents the physics involved in machining based on the thermal hypothesis, and the energy of one pulse turns into heat. Herein, the authors decided the process parameters from the equation that affect the cut-width of PCM. Furthermore, it is also helpful for the reader to conduct a future study on PCMs.
If suggested to remove these equations, it will not affect the content of our findings.
The title says that accuracy is investigated. Therefore, it is necessary to clearly describe and show on the figure exactly which accuracy parameters are being investigated: surface roughness, form error, etc. How is this evaluated using mathematical models?
Response: In this study, the regression equation is developed using response surface methodology. The cut-width (kerf) is investigated in this study. The model is further analyzed for the nonlinear cut (slot) made on the selected PCM (Pl refer Figure 9). Thus, the percentage error obtained is calculated as 5.906%, which is acceptable and shows the competency of the model (attached file).
Small paragraph to figure 10 with a brief description of the conductive coating. Now the details are absolutely incomprehensible. How is the coating applied? Which setting? With what modes, thickness? How will this coating affect the further performance of the resulting product? All this should be clearly, logically described and researched.
Response: The conductive layer of Titanium is applied as shown in figure 5, and it is also mentioned in the manuscript about the thickness of the layer applied on PCM, including other detail is added (highlighted in RED text)
Figure 5: Pictorial view of PCM machining set-up (attached file)
- The authors conclude:
"It was observed that the difference in the values ​​obtained from the theoretical model and the regression model are no more than 15%, thus shows the competence of these models."
However, how does this 15% relate to the accuracy specified by the drawing and by the designer? For example, if the size tolerance is 25 microns, and your model gives a 15% error of 100 microns, then there will be no satisfactory model accuracy.
Response: The contents of the manuscript are significantly CHANGED; the above claim is modified for more clarity for the readers.
Figure 11. Explain in more detail what kind of gear you received? What geometry? Which tooth profile (straight-sided, involute, etc.?) For which application? What is the specified degree of precision of the gear?
Response: The CNC program is designed to identify the correct trajectory/clearances of ET is by using the results obtained in our study i.e. clearance in cut-width. The gear shaped part made by PCM and its possible application and precision of teeth profile is beyond the scope of this paper.
I emphasize that the authors are expected to provide clear and detailed answers to each comment. Appropriate descriptions should be added in the text of the article. Without these changes, the article cannot be accepted for publication.
Response: The CNC program is designed to recognize ET's correct trajectory/clearances by using the results obtained in our study, i.e., clearance in cut-width. The gear-shaped part made by PCM and its possible application and precision of teeth profile is beyond the scope of this paper.
Authors sincerely thank the reviewer(s) for their valuable comments and suggestions that helped to
improve the quality of the paper.
Thanks & Regards

Reviewer 2 Report
Comment 1
Page 1
Polymer Composite Material (PCM) . The conductivity of the workpiece
replace (delete a space)
Polymer Composite Material (PCM). The conductivity of the workpiece
Comment 2
Page 3
Wire-cut electrical discharge
machining (WIRE-EDM ) has been proven to be a potential
replace (Delete a space and insert capital letters)
Wire-cut Electrical Discharge
Machining (WIRE-EDM) has been proven to be a potential
Comment 3
Page 4
There are 47 reference papers, but [49-51]:
the equation [49-51]:
Literature data [31, 51] showed that MRR
(5) [31,50]
Change the numbers of the reference papers.
Comment 4
Page 5
diecelctric fluid; K2
Check the whole text spelling and typographical errors
Delete the space.
Comment 5
Page 5
(as shown in fig 4) can be
replace
(as shown in Figure 4) can be
Comment 6
Page 5
shown in Table 1
replace (Add a .)
shown in Table 1.
Comment 7
Page 6
Section 2.2.
Text full alignment.
There is a different format for the paragragh "The process parameters are presented in Table 2."
Change such as the text has the same format (according to the journal's instructions).
Comment 8
Page 6
Table2. Process
replace
Table 2. Process
Comment 9
Page 7
First the authors mention Table 3 in the text and then Figure 5,
while the order presented in the paper is first Figure 5 and then Table 3.
Move the Table 3 before Figure 5.
Comment 10
Pages 8 - 9
Section 3
Text full alignment.
From table 4,
replace
From Table 4,
(From table 5)
replace
(From Table 5)
Comment 11
Page 11
Figure 9
There is not L3 in the Figure 9 (2 L1).
Comment 12
Page 11
References
The authors must format the text according to the journal's instructions:
[1 - 9], [28 - 31] and [33 - 47]
Author Response
Authors sincerely thank the reviewer(s) for their valuable comments and suggestions that helped to improve the quality of the paper.
Thanks & Regards

Round 3
Reviewer 1 Report
The authors offered satisfactory answers. The article can be accepted for publication.